# Impact of the Visual Performance Reinforcement Technique on Oral Hygiene Knowledge and Practices, Gingival Health, and Plaque Control in Hearing- and Speech-Impaired Adolescents: A Randomized Controlled Trial

**DOI:** 10.3390/children9121905

**Published:** 2022-12-05

**Authors:** Ram Surath Kumar, Apurva Prashant Deshpande, Anil V. Ankola, Roopali M. Sankeshwari, Vinuta Hampiholi, Mamata Hebbal, Sagar Jalihal, Atrey J. Pai Khot, Deepika Valakkunja, Sree Lalita Kotha

**Affiliations:** 1Department of Public Health Dentistry, KLE VK Institute of Dental Sciences, KLE Academy of Higher Education and Research (KLE University), Belagavi 590010, India; 2Department of Periodontics, KLE VK Institute of Dental Sciences, KLE Academy of Higher Education and Research (KLE University), Belagavi 590010, India; 3Department of Preventive Dental Sciences, College of Dentistry, Princess Nourah bint Abdulrahman University, P.O. Box 84428, Riyadh 11671, Saudi Arabia; 4Department of Basic Dental Sciences, College of Dentistry, Princess Nourah bint Abdulrahman University, P.O. Box 84428, Riyadh 11671, Saudi Arabia

**Keywords:** deafness, health education, health knowledge, attitudes, practice, hearing loss, oral health, oral hygiene, sign language, speech disorders

## Abstract

This study aimed to evaluate the impact of oral health education (OHE), incorporating a novel pre-validated visual performance reinforcement (VPR) technique and sign language, on gingival health, plaque control, and oral hygiene knowledge and practices in 12 to 15-year-old hearing- and speech-impaired adolescents. A double-blinded randomized controlled trial was conducted in a government school for deaf children in Belagavi, Karnataka, India. A total of 80 adolescents, aged 12–15 years, were randomly assigned, using a computer-generated table of random numbers, into two groups: Group A receiving the VPR technique (*n* = 40), and Group B receiving sign language (*n* = 40). A specially designed pre-validated closed-ended questionnaire was administered to both groups, followed by clinical examination to obtain the gingival and plaque index, before intervention and at a 16-week follow-up period. Group A showed a significant increase in the knowledge gained when compared to Group B. Similarly, a significant improvement in oral hygiene practices was also observed in Group A. However, at the 16-week follow-up, there were no statistically significant differences in gingival and plaque scores between the groups. OHE using the VPR technique can be as effective and satisfactory as sign language in the reduction of gingival and plaque scores and in the improvement of knowledge and its application in oral hygiene maintenance among hearing- and speech-impaired adolescents.

## 1. Introduction

Hearing and speech impairment are the most prevalent sensory deficits [1]. According to the World Health Organization (WHO), in 2022, there are 430 million people worldwide with debilitating hearing loss, of which 34 million are children [2]. Asia Pacific, South Asia, and Sub-Saharan Africa are the primary regions burdened with debilitating hearing loss, with a prevalence rate four times higher than that of high-income regions [3]. In India, more than 3 million people suffer from debilitating hearing loss, and over 1.2 million people suffer from speech impairment [4]. According to the National Sample Survey Organization of India 2018, the disability incidence was 86 per 100,000 people. This prevalence accounts for 0.5% of hearing and speech deficits [5].

Hearing and speech impairments are both significant concerns, since they impede growth and development and lower the quality of life of an individual. Communication barriers develop as a consequence of hearing and speech impairment, resulting in mental distress, physical and emotional abuse, and poor social relationships, contributing to limited and poor maintenance of oral health. Communication through visual displays and illustrations breaks the barriers and helps in better understanding and implementation in general [6,7,8,9,10].

A study conducted by Vignehsa et al. [11] in Singapore reported a high prevalence of gingivitis (98%) and dental caries (82%) among hearing- and speech-impaired children, and they also received less dental attention. Similarly, Indian studies reported a high prevalence of dental caries and gingivitis, 65% and 47%, respectively, among hearing- and speech-impaired children [12]. Hence, a need for non-discriminatory comprehensive health care services is emphasized. Primary deterrence is crucial in the prevention of oral diseases in any population, including children with developmental disabilities. As a result, it is essential to communicate with them and educate them on the importance of oral hygiene. Dental health professionals observed an impediment in communication and motivation when educating children regarding various life skills, including the maintenance of oral hygiene [13].

Sign language is often considered as the gold standard for communicating and assisting hearing impaired children in acquiring literacy skills. However, it necessitates the support of trained personnel for effective interaction. This limitation can be overcome by an innovative method of delivering oral health education (OHE) using the visual performance reinforcement (VPR) approach. The present study used a VPR technique where OHE, in the form of a customized animated visual playback with cartoon characters, provided OHE in sign language with subtitles in English, which included a demonstration of the brushing technique on a dental model, followed by the periodic reinforcement of visual display and demonstration. This technique could be a breakthrough in the arena of OHE, especially for hearing- and speech-impaired adolescents. Hence, the aim of the study was to assess the effectiveness of the VPR technique and sign language on oral health among hearing- and speech-impaired adolescents, aged 12–15 years. Therefore, with a 16-week intervention, we tested the hypothesis that there is a difference in oral hygiene knowledge and practices, gingival health, and plaque control after imparting OHE using VPR technique and sign language among hearing- and speech-impaired adolescents.

## 2. Materials and Methods

### 2.1. Study Design and Study Setting

The study was carried out as a double-arm double-blinded randomized controlled trial in a government school for deaf children in Belagavi, Karnataka, India, from March to June 2022. The trial was registered under the Clinical Trials Registry—India, with the CTRI number CTRI/2022/01/039703. Consolidated Standards of Reporting Trials (CONSORT) guidelines were followed.

### 2.2. Sample Size Calculation

The sample size for the study was calculated using the GPower program (G * Power Version 3.1.9.4 statistical software). The sample size was estimated to be 35 adolescents in each group, accounting for a total sample size of 70 at a power of 0.85 and an alpha error of 0.05 [13]. Hence, taking into consideration a 10% dropout rate, this study included a sample size of 80 adolescents in total, with 40 adolescents in each group.

### 2.3. Inclusion and Exclusion Criteria

#### 2.3.1. Inclusion Criteria

(1)Participants in the age group of 12–15-years old, possessing hearing loss with speech impairment.(2)Those who were not utilizing hearing aids.(3)Those who were trained in sign language.(4)Those having a plaque score greater than one.

#### 2.3.2. Exclusion Criteria

(1)Participants with other disabilities, in addition to hearing and speech impairment.(2)Participants and their parents/guardians who did not provide assent and written informed consent to participate in the study.(3)Those with uncooperative behavior.(4)Those who were unable to cope with the oral examination procedure.(5)Those undergoing orthodontic treatment and/or any other dental treatment in the previous six months.(6)Those with underlying systemic disease.(7)Those with special health care needs.(8)Those using medications that could affect gingival health.

### 2.4. Ethical Considerations

The ethical clearance was obtained from the Institutional Research and Ethics Committee with reference number 1415, dated 14 March 2021. This study strictly adhered to the ethical standards of human experimentation and the Helsinki Declaration of 1975, as revised in 2000.

### 2.5. Assessment Plan 

#### 2.5.1. Preparatory Phase

The investigator enrolled in a three-month certified training program to learn sign language. Two examiners were trained to record gingival and plaque indices and were supervised by subject experts. Intra-examiner and inter-examiner reliability were calculated (0.81, 0.85) and (0.83, 0.88), respectively, using kappa statistics, which indicated a substantial level of agreement. A specially designed pre-validated 17-item closed-ended questionnaire was prepared. The face and content validity of the study was confirmed by five subject experts in order to determine the validity. Cronbach’s alpha was utilized to assess the validity and internal consistency of the closed-ended questionnaire using the content validity ratio (0.85) from a pilot study conducted on a group of twelve hearing and speech impairment adolescents between the ages of 12 to 15 years. The pilot study was carried out to check the reliability and validity of the questionnaire, the feasibility of the study, and the response of the participants to the visual aids. Participants requested a reduction in the speed of the visual content to perceive it better, and the adjustment was carried out based on their feedback. The pilot study results were not included in the main study.

#### 2.5.2. Pre-Education and Randomization

With the help of school teachers, interactive sessions with the hearing- and speech-impaired adolescents were conducted to gauge the level of cognition and cooperation.

A total of 80 volunteering adolescents were included in the study using a simple random sampling technique. Participants were randomly assigned into two groups using a computer-generated table of random numbers, Group A receiving the VPR technique (*n* = 40) and Group B receiving sign language (*n* = 40). Allocation concealment was performed using the SNOSE (sequentially numbered, opaque, sealed envelope) technique. The questionnaire, in the English language, was distributed, and the responses, including information such as sociodemographic details and information on oral hygiene knowledge and practices, were collected by the examiners. This was followed by the clinical examination and recording of gingival and plaque scores by the examiners using the Loe and Silness gingival index [14] and the Silness and Loe plaque index [15]. The type of intervention provided by the investigator was masked from the examiners, and these results were also blinded regarding group assignment. 

#### 2.5.3. Administration of Intervention

After the baseline assessment, the primary investigator delivered OHE using the VPR technique and sign language to Group A and Group B participants, respectively.

*VPR Technique:* A customized pre-validated animated video, with cartoon characters providing OHE in sign language with subtitles in English for the duration of 9:09 min, was projected onto a screen to the participants in Group A. In the video, the cartoon characters were depicted performing sign language to deliver OHE. The display of these cartoon characters was created with the help of the Hand Talk software application (https://apps.apple.com/in/app/hand-talk/id659816995 (accessed on 8 July 2021)). The visual content consisted of an introduction to oral health and its significance in general health (3:10 min), brushing technique (3:51 min), and the golden rules for maintaining effective oral health (2:08 min). In the next phase of the study, the Modified Bass brushing technique was demonstrated by the investigator on a dental model. Following this, each participant was encouraged to perform the brushing technique on the dental model until they achieved perfection in a practice session of four minutes under the supervision of the investigator. Finally, periodic reinforcement of OHE was provided at 1-week, 4-week, and 8-week intervals to ten participants as a group.

*Sign Language Technique:* Participants in Group B attended a 20 min OHE given by an investigator trained in sign language after receiving the approval of sign language experts and school teachers for a smooth relay of OHE. The content of the health education covered was similar to that of Group A, followed by the demonstration of the Modified Bass brushing technique. A CONSORT flow diagram is shown in Figure 1.

### 2.6. Follow-Up

The gingival and plaque scores of all the participants in both groups were recorded at the 16th week, using the same indices. The same questionnaire was administered to assess oral hygiene knowledge and practices after 16 weeks to estimate the impact of both OHE techniques among the participants. To minimize investigator bias, the clinical examination was carried out by the same examiners, who were blinded to group assignment. The participants were unaware of the exact dates of the intervention and assessment. The content of health education was similar in both groups. 

To fulfil ethical obligations, the VPR technique was presented to the participants of Group B as well, after the completion of the study.

The questionnaire consisted of 17 items of which 13 were knowledge-based, to be answered by 40 participants of each group and were scored from 0 to 13. The correct response was scored as “1” and the incorrect response as “0”. The total knowledge score was computed based on the response of each participant. The overall score was totaled by a simple sum of responses.

### 2.7. Statistical Analysis

Data obtained were entered in Microsoft Excel 2020, subjected to statistical analysis by a blinded statistician, and analyzed using the IBM Corp. Released 2012, IBM SPSS^®^ Statistics for Windows, Version 21.0. Armonk, NY: IBM Corp. The descriptive statistics were presented as mean ± standard deviation for continuous variables and as frequencies with percentages for categorical variables. The normality of the distribution of the continuous variable was determined using the Shapiro–Wilk test. As the data were normally distributed *(p* > 0.05), Chi-square and McNemar analyses were carried out to analyze the differences in the responses, before and after OHE intervention, in both groups. Paired and unpaired *t*-tests were applied to compare the mean gingival score, plaque score, and knowledge score within and between Groups A and B, respectively. Drop-out analysis was performed using *t*-tests [16]. Statistical significance was set at *p* ≤ 0.05.

## 3. Results

The mean age of the participants in Group A and Group B was 13.22 ± 0.74 and 13.76 ± 0.91 years, respectively. Of the total sample of 80, the ratio of girls (65%) was greater than boys (35%) in both groups. Table 1 and Table 2 summarize the oral hygiene knowledge and practices, respectively, in the study population of both groups, before and after OHE intervention. The unpaired *t*-test revealed that the baseline knowledge score in both groups was almost equal and statistically insignificant (*p* = 0.982). The McNemar test showed that the percentage of correct answers was significantly higher after the intervention (*p* < 0.05). The mean knowledge score at the 16-week follow-up indicated a statistically significant difference between the groups, *p* < 0.001. Following the intervention, Group A showed a significant increase in the knowledge gained (score: 5.71 ± 1.64) when compared to Group B (score: 3.54 ± 1.71) while using the unpaired *t*-test, *p* < 0.001 (Table 3). Similarly, significant improvement in oral hygiene practices was also observed in Group A when compared to Group B (Table 2).

Figure 2 summarizes the mean gingival and plaque scores, before and after OHE intervention. The unpaired *t*-test revealed the baseline gingival and plaque scores in both groups to be almost equal and statistically insignificant (*p* > 0.05). The paired *t*-test revealed that both indices indicated a statistically significant reduction in the gingival and plaque scores in both groups from baseline to the 16-week interval (*p* < 0.001). However, at the 16-week follow-up, the gingival scores in Group A and Group B were 0.33 ± 0.24 and 0.29 ± 0.25, respectively. The plaque scores between Group A and Group B were 0.58 ± 0.33 and 0.53 ± 0.23, respectively, with no statistically significant differences between the groups using the unpaired *t*-test (*p* > 0.05). However, the dropout analysis ensured that the present study was adequately powered, as the dropouts and completers did not differ significantly, thereby avoiding the differential bias.

## 4. Discussion

Adolescents with hearing and speech impairment encounter cognitive deficits, such as delayed language and knowledge acquisition, communication difficulties, social isolation, and stigmatization, thus undesirably impacting personal health. According to Roland et al., a negative correlation exists between hearing impairment and quality of life [17]. Communication is a two-way process. However, these children are deprived of good oral health due to communication barriers. They require assistance for understanding, as well as training for effective communication [7]. Oral health promotion can be accomplished effectively if the requirements of this special population are fulfilled.

Flanders stated that when special measures and programs are implemented for children, oral hygiene tends to improve greatly [18]. The ability of the children to pay attention, retain information, and recall from memory under controlled circumstances is essential for the success of any educational method [19]. In most cases, children with special needs tend to learn by imitating or replicating; therefore, it is essential to incorporate visual aids suitable for their level of cognitive ability to help them retain the knowledge and implement it by mimicking it in their day-to-day life.

Sign language has been routinely employed to assist these children in acquiring skills needed for reading and writing [20]. The same approach can be employed when demonstrating oral hygiene behaviors and practices to these children. In sign language, words are symbolized by making different shapes with the fingers and hands, each representing a distinct alphabet, which requires practice, effort and expertise [21]. Sign language necessitates the support of trained personnel for effective interaction. Dental health professionals need additional training to learn sign language in order to deliver effective OHE. This limitation can be overcome by the VPR technique. Hence, in the present study, the VPR technique was employed to deliver OHE to hearing- and speech-impaired adolescents. This technique is based on the social learning theory, which states that the majority of behaviors of an individual are acquired through personal experience or by observations [19].

The overall study findings demonstrate a significant improvement in the knowledge and practical behavior regarding oral hygiene maintenance following the OHE intervention in both groups. This indicates that the children understood the relevance of oral health to general health, as well as brushing technique and the golden rules for maintaining effective oral hygiene. The VPR technique group showed a significant knowledge gain compared to the sign language group. The study conducted by Fageeh et al. reported similar findings of improved oral health knowledge and hygiene practices after following the oral hygiene instructions portrayed in the videos using Arabic sign language, along with captions [22].

The baseline plaque score in both groups was fair, denoting a lack of awareness among hearing- and speech-impaired children, which is in accordance with the studies conducted by Sandeep et al. [23], Doichinova et al. [24], and Shetty et al. [25]. However, the studies conducted by Hashmi et al. [13] and Pareek et al. [26] showed sign language as the major factor in improving oral health as compared to conventional OHE. At the end of 16 weeks in the present study, both groups indicated a significant reduction in gingival and plaque scores when compared with the baseline. This may be attributed to the knowledge imparted by the health education methods that allowed the children to comprehend, absorb, and adopt them into their routine oral hygiene habits.

It has been suggested that hearing- and speech-impaired children rely heavily on visual modality to acquire knowledge [27]. In this study, the use of animated visual aids involving cartoon characters imparting OHE by sign language, as well as the demonstration of the Modified Bass brushing technique, helped improve oral health status, with a significant reduction in gingival and plaque scores in the VPR group. Furthermore, multiple reinforcements may have impacted the retention of the knowledge gained, thus playing a key role in their motivation. Emier et al. confirmed that periodic reinforcements can promote better oral hygiene in children [28]. 

Although there was a significant reduction in gingival and plaque scores when compared with the baseline, there was no significant difference in gingival and plaque scores between the groups at the end of 16 weeks. These findings could be attributed to the efficiency of both the VPR and sign language techniques. The study conducted by Baliga et al. [29] reported that both video modeling and sign language equivalently impacted the oral health status of hearing-impaired children, which is in accordance with the findings of the present study.

### 4.1. Strength and Limitations

The selected participants were resident adolescents living at the school, who were provided with and consumed the same diet. The study was balanced by selecting participants with baseline plaque scores greater than one. 

This novel approach of imparting OHE using the VPR technique provides alternative effective intervention by health care professionals, enhancing interest and building rapport with the children. However, the limitation of the study is the restricted follow-up period.

This study was the first of its kind to employ this novel VPR technique to incorporate visuals, demonstration, and reinforcement to create an engaging and entertaining medium of instruction for OHE to a specific population to create a positive influence on the knowledge and practices of oral hygiene habits. This strategy can be adopted as an OHE tool, with the added advantage of being capable of repetitive employment, with no additional expenses or efforts to train health professionals in sign language.

### 4.2. Future Prospects and Recommendations

To substantiate the findings of this study, longitudinal studies encompassing multiple health education sessions involving children, teachers, parents, and health care professionals in a larger population should be conducted. Such findings can be extrapolated to children of all ages, socioeconomic backgrounds, and schools in various geographical regions.

### 4.3. Clinical Significance 

Specific OHE techniques catering to populations with special needs have a long-lasting impact on our society. Hence, the VPR technique can be used as a fundamental health education tool to increase awareness among hearing- and speech-impaired adolescents. It is a simple, effective alternative which employs fun tactics of visual animation and an engaging projection to provide oral hygiene instructions, and it is thus recommended to all healthcare professionals.

## 5. Conclusions

OHE using the VPR technique is as effective as sign language in the reduction of gingival and plaque scores and in the improvement of knowledge and its application in oral hygiene maintenance among hearing- and speech-impaired adolescents. The incorporation of the VPR technique in OHE programs is an easy, engaging, and cost-effective method. This technique, visual performance reinforcement, could be a breakthrough in the arena of OHE, especially when incorporated with conventional OHE for hearing- and speech-impaired adolescents. VPR, being a novel technique, can be reformed and utilized for hearing- and speech-impaired children of all ages for imparting OHE and for reaching out to inaccessible and/or underprivileged areas. In the future, this can be an important tool to act as a medium of instruction in all schools, leading to a dramatic improvement in the oral health of generations to come.

## Figures and Tables

**Figure 1 children-09-01905-f001:**
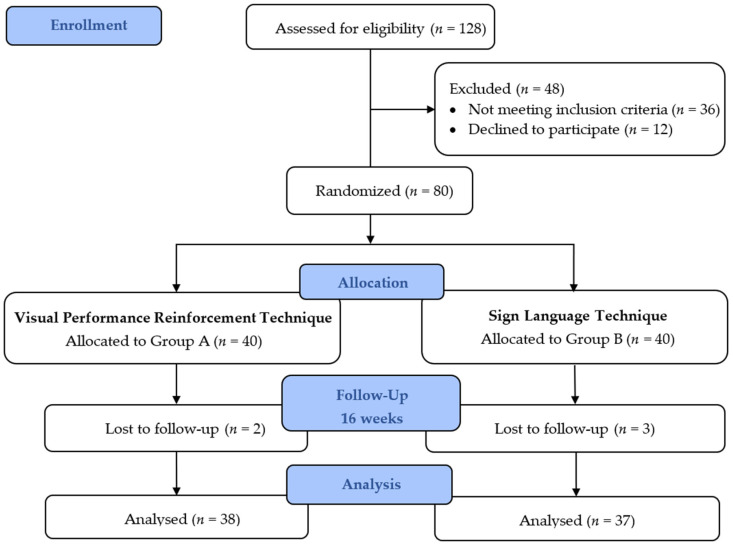
Consolidated Standards of Reporting Trials (CONSORT) diagram.

**Figure 2 children-09-01905-f002:**
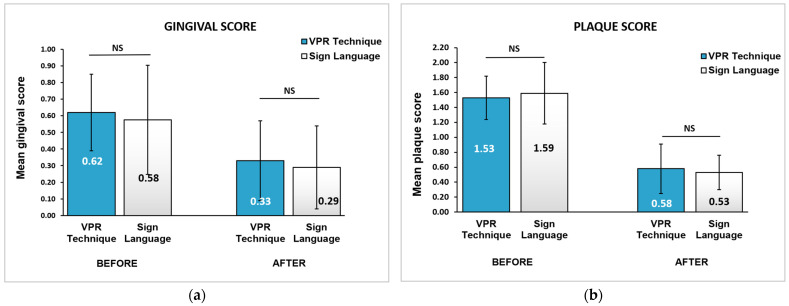
Comparison of (**a**) mean gingival score and (**b**) mean plaque score in the VPR technique group and the sign language group. All values are expressed as mean ± standard deviation. The statistical test used: unpaired *t*-test; Statistical significance was set at *p* ≤ 0.05; NS: not significant.

**Table 1 children-09-01905-t001:** Comparison of oral hygiene knowledge in the study population before and after the VPR technique and sign language technique intervention.

Questions		Response	VPR Technique (*n* = 38)		Statistics	Sign Language (*n* = 37)		Statistics
			Before	After	*p*-Value	Before	After	*p*-Value
1.	Does oral hygiene have any role in the maintenance of general health?	Yes ^α^	27 (71.1%)	36 (94.7%)	0.012 *	26 (70.3%)	30 (81.1%)	0.454
		No ^β^	11 (28.9%)	2 (5.3%)		11 (29.7%)	7 (18.9%)	
2.	Which among the two is the best tooth cleaning agent?	Toothpaste ^α^	26 (68.4%)	36 (94.7%)	0.006 *	30 (81.1%)	32 (86.5%)	0.754
		Toothpowder ^β^	12 (31.6%)	2 (5.3%)		7 (19.9%)	5 (13.5%)	
3.	How long should you brush your teeth?	2 to 3 min ^α^	6 (15.8%)	21 (55.3%)	0.001 **	11 (29.7%)	32 (86.5%)	0.001 **
		45 s ^β^	32 (84.2%)	17 (44.7%)		26 (70.3%)	5 (13.5%)	
4.	How many times should you brush your teeth in a day?	Twice a day ^α^	2 (5.3%)	30 (78.9%)	0.001 **	1 (2.7%)	10 (27.0%)	0.012 *
		Once in a day ^β^ Once in two days ^β^	36 (94.7%)	8 (21.1%)		36 (97.3%)	27 (73.0%)	
5.	Which is the best time to brush your teeth?	Morning and night ^α^	23 (60.5%)	32 (84.2%)	0.001 **	20 (54.1%)	36 (97.3%)	<0.001 **
		Only morning ^β^ Only night ^β^	15 (39.5%)	6 (15.8%)		17 (45.9%)	1 (2.7%)	
6.	How do you clean your teeth regularly?	Brushing with toothpaste, toothbrush and rinsing after meal ^α^	4 (10.5%)	32 (84.2%)	<0.001 **	3 (8.1%)	7 (18.9%)	<0.001 **
		Brushing with finger ^β^ Brushing with toothpaste and toothbrush ^β^	34 (89.5%)	6 (15.8%)		34 (91.9%)	30 (81.1%)	
7.	When do you have to change your toothbrush?	Once every three months ^α^	6 (15.8%)	34 (89.5%)	<0.001 **	2 (5.4%)	32 (86.5%)	<0.001 **
		Once a month ^β^ Once every two months ^β^ Once every six months ^β^	32 (84.2%)	4 (10.5%)		35 (94.6%)	5 (13.5%)	
8.	Which brushing technique is the best to clean your teeth?	Gentle downward and upward strokes, along with a circular motion ^α^	7 (18.4%)	30 (78.9%)	<0.001 **	9 (24.3%)	32 (86.5%)	<0.001 **
		Horizontal scrub motion ^β^	31 (81.6%)	8 (21.1%)		28 (75.7%)	5 (13.5%)	
9.	What will happen if you do not brush your teeth regularly?	Both of the above ^α^	16 (42.1%)	30 (78.9%)	0.004 *	18 (48.6%)	24 (64.9%)	0.263
		Tooth decay causing tooth loss ^β^ Gum disease causing tooth loss ^β^ I don’t know ^β^	22 (57.9%)	8 (21.1%)		19 (51.4%)	13 (35.1%)	
10.	You should clean your teeth after every meal.	True ^α^	30 (78.9%)	36 (94.7%)	0.070	32 (86.5%)	36 (97.3%)	0.125
		False ^β^	8 (21.1%)	2 (5.3%)		5 (13.5%)	1 (2.7%)	
11.	Sweet and sticky foods containing sugar are healthy for your teeth.	False ^α^	22 (57.9%)	30 (78.9%)	0.096	22 (59.5%)	27 (73.0%)	0.359
		True ^β^	16 (42.1%)	8 (21.1%)		15 (40.5%)	10 (27.0%)	
12.	Periodic check-up visits to a dentist are important to maintain the health of your mouth.	True ^α^	25 (65.8%)	38 (100%)	-	22 (59.5%)	32 (86.5%)	0.021 *
		False ^β^	13 (34.2%)	0 (0%)		15 (40.5%)	5 (13.5%)	
13.	How often should you visit the dentist?	Once every 6 months ^α^	5 (13.2%)	32 (84.2%)	<0.001 **	5 (13.5%)	16 (43.2%)	0.013 *
		Once a month ^β^ Once every 3 months ^β^ Once a year ^β^	33 (86.8%)	6 (15.8%)		32 (86.5%)	21 (56.8%)	

VPR technique: visual performance reinforcement technique. All values are expressed as the frequency with percentages (in parentheses); ^α^ denotes correct response, and ^β^ denotes incorrect response. The statistical test used: McNemar test; level of significance: * *p* ≤ 0.05 is considered statistically significant; ** *p* ≤ 0.001 is considered a highly significant association.

**Table 2 children-09-01905-t002:** Comparison of oral hygiene practices in the study population, before and after the intervention with the VPR technique and the sign language technique.

Questions		Response Frequencies *n* (%)						
		Response	VPR Technique (*n* = 38)		Statistics	Sign Language (*n* = 37)		Statistics
			Before	After	*p*-Value	Before	After	*p*-Value
1.	Do you use fluoride-containing toothpaste?	Yes ^α^	5 (13.2%)	22 (57.9%)	<0.001 ^§,^**	4 (10.8%)	19 (51.4%)	<0.001 ^§,^**
		No ^β^	33 (86.8%)	16 (42.1%)		33 (89.2%)	18 (48.6%)	
2.	Do you use dental floss?	Yes ^α^	0 (0%)	20 (52.6%)	-	1 (2.7%)	11 (29.7%)	0.006 ^§,^*
		No ^β^	38 (100%)	18 (47.4%)		36 (97.3%)	26 (70.3%)	
3.	Do you clean your tongue?	Yes ^α^	4 (10.5%)	24 (63.2%)	<0.001 ^§,^**	7 (18.9%)	16 (43.2%)	0.004 ^§,^*
		No ^β^	34 (89.5%)	14 (36.8%)		30 (81.1%)	21 (56.8%)	
4.	Where did you get this information?	Parents/guardians	7 (%)	0 (%)	0.001 ^||,^**	6 (%)	0 (0%)	0.001 ^||,^**
		Teachers	16 (%)	5 (%)		13 (%)	9 (0%)	
		Dentist	3 (%)	0 (%)		1 (%)	0 (0%)	
		Have not received any information on this *(before intervention)*	12 (%)	-		17 (0%)	-	
		From this health education module *(after intervention)*		33 (%)			28 (%)	

VPR technique: visual performance reinforcement technique; OHE: oral health education. All values are expressed as frequencies with percentages (in parentheses); ^α^ denotes good practice, and ^β^ denotes bad practice. The statistical test used: ^§^ McNemar test and ^||^ Chi-square test; level of significance: * *p* ≤ 0.05 is considered statistically significant; ** *p* ≤ 0.001 is considered highly significant association.

**Table 3 children-09-01905-t003:** Comparison of oral hygiene knowledge score in the study population of VPR technique and sign language technique before and after the intervention.

Knowledge Score	VPR Technique (*n* = 38)		Sign Language (*n* = 37)		Statistics	
	Mean ± SD	95% CI	Mean ± SD	95% CI	*t*-Value	*p*-Value ^||^
**Before intervention**	5.26 ± 1.22	4.81–5.62	5.49 ± 1.22	5.08–5.89	−0.023	0.982
**After intervention**	10.97 ± 1.31	10.56–11.44	9.03 ± 1.57	8.50–9.55	5.842	<0.001 **
***t*-value**	21.432		12.599			
***p*-value ^§^**	<0.001 **		<0.001 **			
**Knowledge gain**	5.71 ± 1.64	5.17–6.25	3.54 ± 1.71	2.97–4.11		<0.001 **

VPR technique: Visual performance reinforcement technique; CI: confidence interval. All values are expressed as mean ± standard deviation (SD). The statistical test used: ^§^ Paired *t*-test, ^||^ Unpaired *t*-test; Level of significance: ** *p* ≤ 0.001 is considered highly statistically significant.

## Data Availability

Data available on request, to maintain confidentiality. The data presented in this study are available on request from P.I. (first author). The data are not publicly available due to detailed information about the participants contained in the data.

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
