# Peer review of "Impact of the Visual Performance Reinforcement Technique on Oral Hygiene Knowledge and Practices, Gingival Health, and Plaque Control in Hearing- and Speech-Impaired Adolescents: A Randomized Controlled Trial"

_children, 2022, doi:10.3390/children9121905_

Round 1

Reviewer 1 Report

Dear authors,

Impact of Visual Performance Reinforcement Technique and Sign Language Technique on Gingival Health, Plaque Control, Oral Hygiene Knowledge and Practices among Hearing and Speech Impaired Adolescents: A Randomized Controlled Trial

The title is too long, consider shortening it

Abstract: if at 16 weeks of follow-up, there were no statistically significant differences in gingival and plaque scores between the group, what is the clinical relevance?

Keywords: please use Mesh terms

Introduction: Indian studies reported a high prevalence of dental caries and gingivitis among hearing and speech-impaired children – please consider other parts of the globe as well

The aim of the study should be better emphasized. No need of separating between null hypothesis is recommended

The inclusion and exclusion criteria should be better described – more in detail 

The validation of the closed-ended questionnaire should be described/ what do you mean by closed – ended?

The pilot study should be explained – line 112-118

Material and methods

The movie and the other methods should be provided as supplementary files

The total knowledge score was computed based on the response of each participant. 169 The correct response was scored as “1” and the wrong response as “0”. The overall score 170 was a simple sum of responses ranging from 1 to 13. – is unclear – please explain

The conclusion should be better emphasized 

The Logistics Rule: 3W (what / where / when) for 1 H (how) – should be followed 

 References should be in journal style

Reviewer 2 Report

The study is well organized. The results are consistent with the study design.

It meets the methodological requirements of an RCT.

Reviewer 3 Report

Interesting and very important topic. Methodology is well designed.  As a sensory deficit is very prevalent nowadays these paper have huge social  importance. It should be emphasize  in paper. Also I suggest to add detailed description of VPR approach in introduction section. My suggestion is to add more detailed description about clinical indexes, and explanation why Bleeding on probing index was not included, besides PI and GI. 

Round 2

Reviewer 1 Report

Congratulations on your work!